# Does knowledge of the primary tumour affect survival after surgery for spinal metastatic disease? A retrospective longitudinal cohort study

Christian Carrwik ,[1] Claes Olerud,[1] Yohan Robinson [1,2,3]

[1]Department of Surgical Sciences, Uppsala University, Uppsala, Sweden
[2]Department of Research and Development, Armed Forces Centre for Defence Medicine, Gothenburg, Sweden
[3]Institute of Clinical Sciences, Sahlgrenska Academy, Gothenburg University, Gothenburg, Sweden

**Correspondence to**
Dr Christian Carrwik;
christian.carrwik@surgsci.uu.se

## ABSTRACT

**Objectives** To compare survival after surgery for patients with spinal metastatic disease with known primary tumour (KPT) versus patients with unknown primary tumour (UPT).

**Participants** 393 patients 18 years or older (270 men and 123 women, mean age 67.3 years) undergoing surgery at Uppsala University Hospital in Sweden between 2006 and 2016 due to spinal metastatic disease . 271 patients (69%) had a KPT at the time of surgery and 122 (31%) had an UPT.

**Interventions** Decompressive and/or stabilising spine surgery due to spinal metastatic disease.

**Primary outcome** Survival (median and mean) after surgery.

**Results** The estimated median survival time after surgery for patients with KPT was 7.4 months (95% CI 6.0 to 8.7) and mean survival time was 21.6 months (95% CI 17.2 to 26.0). For patients with UPT, the median estimated survival time after surgery was 15.6 months (95% CI 7.5 to 23.7) and the mean survival time was 48.1 months (95% CI 37.3 to 59.0) (Breslow, p=0.001). Unknown primary cancer was a positive predictor of survival after surgery (Cox regression, HR=0.58, 95% CI 0.46 to 0.73).

**Conclusion** In this study, patients with spinal metastasis and UPT had a longer expected survival after surgery compared with patients with KPT. This suggests that patients with UPT and spinal metastasis should not be withheld from surgery only based on the fact that the primary tumour is unknown.

## Strengths and limitations of this study

► Large single-centre cohort with at least 4 years follow-up time.
► Reliable survival data with linkage to the Swedish Population Register.
► Includes results of biopsies of the unknown primary tumours.
► No control group with patients treated only non-surgically.
► Lacks reliable data on quality of life and neurological outcome after surgery.

## INTRODUCTION

The evolution of oncological treatment and longer expected survival has lowered the threshold for surgical intervention among patients with metastatic spine disease in recent decades. In several studies, decompressive surgery (often in combination with surgical stabilisation) has been found to be beneficial for patients with an expected survival of at least 3 months.[1 2] In a recent study by Dea *et al*, this time frame has been challenged and the authors suggest that the extent of the surgery and patient-specific factors rather than expected survival should be guiding in the decision-making process.[3]

The prognosis for the individual patient depends on several factors, including the patient's performance status and the extent of generalisation of the malignancy. One of the most important factors is the type of primary tumour and the underlying prognosis for that specific disease. Most, if not all, prognostic algorithms include some sort of classification of the primary tumour.[4–9] Therefore, the type of primary tumour—and the associated prognosis—is a key factor in selecting treatment, ranging from best supportive care to extensive spinal surgery.

The prognostication becomes even more challenging when the type of primary tumour is unknown, as in the cases when metastatic spine disease is the initial manifestation of malignancy (IMM). When the patient presents with symptoms of epidural spinal cord compression, there is a limited time for further investigations before a decision of treatment. As the duration of neurological impairment is one of the factors predicting the neurological outcome, surgery (if applicable) should take place as soon as possible under the safest possible circumstances.[10]

The frequency of metastatic spine disease as IMM in surgically treated cohorts varies between different studies. In the widely cited randomised controlled trial by Patchell *et*

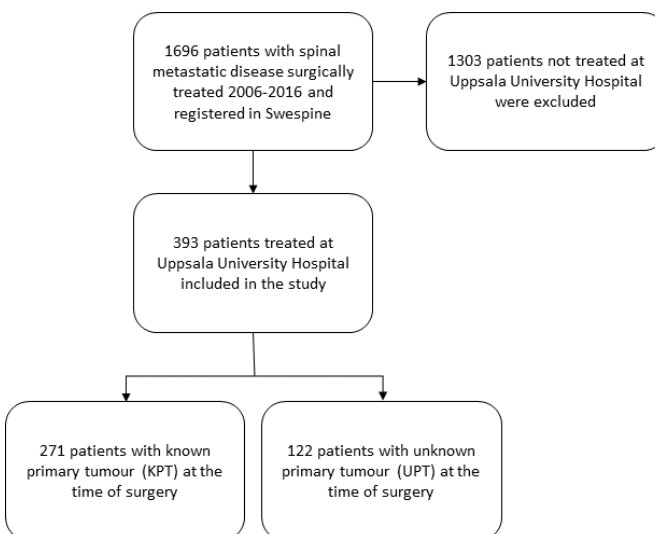

**Figure 1** Flow-chart for inclusion and exclusion in the study.

*al*, 5 out of 50 surgically treated patients (10%) had an unknown primary tumour (UPT). The AO Spine study published in 2016 included 142 surgically treated patients and 17 of those (12 %) had UPT. In a more recent study by Wang *et al* with 448 surgically treated patients during a period of 17 years, 15% of the patients presented with UPT. Other studies report up to 20% of UPT among patients treated surgically.[1 2 11]

Previous studies show varying results regarding survival after surgery for spinal metastatic disease for patients with UPT compared with known primary tumour (KPT). A study published by Quraishi *et al* in 2014, retrospectively evaluating a cohort of 285 patients of which 17 had UPT, reported no significant difference in survival between the two groups but a 'trend towards shorter survival' in the UPT group.[12] A more recent study from Park *et al* showed that patients with spinal metastasis as IMM had longer survival than patients with KPT, but the cases span from a period of 20 years.[13] In the study by Wang *et al* from 2012, the UPT group had a longer median survival than the cohort as a whole (11.4 months vs 6.8 months).[14]

Considering these somewhat conflicting results, we decided to perform a survival analysis of a cohort of patients treated surgically due to spinal metastatic disease at Uppsala University Hospital between 2006 and 2016. A part of this cohort has been investigated earlier by this

group in a study analysing the precision of predictive scores.[15]

## MATERIALS AND METHODS
### Study design
This is a longitudinal retrospective cohort study of consecutively treated patients with spinal metastatic disease.

### Setting
The study was performed in Sweden, where virtually all patients with cancer are treated in the public healthcare. Uppsala University Hospital is one of seven university hospitals in Sweden and referral centre for about 2.1 million people.

Swespine, a national registry for spine surgery, is maintained to monitor and develop the quality of spine surgery clinics. Swespine was established in 1993 and has a coverage of 49 of 50 clinics performing spine surgery in Sweden. A module for spinal metastatic disease was added to Swespine in 2006.

### Participants
Patients 18 years or older undergoing surgery due to spinal metastatic disease at Uppsala University Hospital between 1 January 2006 and 31 December 2016 and registered in Swespine.

Individual patient history, including data on the primary tumour, was obtained from the medical records at Uppsala University Hospital. The records are linked to the Swedish Population Registry, containing data on date of death if applicable.

### Patient and public involvement
Patients or the public were not involved in the design, or conduct, or reporting, or dissemination plans of our research.

### Variables
#### Survival
Survival after surgery is defined as the time difference from the date of surgery, as reported to Swespine, and the date of death as recorded in the Swedish Population Register. All patients defined as alive in this study were alive as of 25 January 2021.

### Type of tumour
Data on primary tumour is entered into Swespine by the surgeon. Available choices are known/unknown primary tumour and if known is chosen, a subset of eight alternatives becomes available. Data on tumour type in cases with UPT is derived from the histopathology report, filed in the medical records, on the tissue sample collected during surgery. As a quality measure, cases registered in Swespine as UPT were verified with the medical records to eliminate 'false positive' UPT cases.

### Indications and types of surgery
In Swespine, the surgeon reports the indication for surgery as well as the type of surgery. Possible indications

| Table 1 | Patient characteristics | | |
|---|---|---|---|
| **Data point** | **Known primary tumour (n=271)** | **Unknown primary tumour (n=122)** | **P value** |
| Mean age, years | 67.5 | 66.9 | 0.47 |
| Male sex | 196 (72%) | 74 (61%) | 0.02 |
| Active smoker (yes/ no) | 11/139 | 9/61 | – |

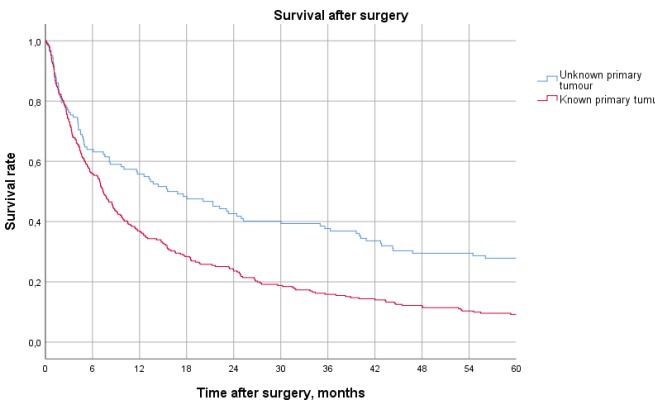

**Figure 2** Kaplan-Meier chart comparing survival in the unknown primary tumour and known primary tumour groups.

## DATA SOURCES/MEASUREMENT

A data set was provided from Swespine including all patients undergoing surgery due to spinal metastatic disease in Sweden between 2006 and 2016. Patients coded as treated at Uppsala University Hospital were selected for inclusion in the study. The data were cross-referenced with the medical records and the Swedish Population Register, using the personal identification number (PIN) for each record (figure 1).

## Bias

There is most probably a selection bias as patients who are selected to undergo surgery generally have a better performance status and/or prognosis than patients selected to undergo non-surgical treatment. On the other hand, the risk for selection bias regarding registration in Swespine and thus inclusion in this study is negligible as

registration in quality registries is regulated by law and is not at the discretion of the patient or the clinician.

## Study size

The study size was determined by the number of patients with spinal metastatic disease being treated at Uppsala University Hospital and registered in Swespine between 2006 and 2016. No power calculations were made prior to the study.

## Statistical methods

IBM SPSS Statistics V.26 was used for all statistical calculations. The survival analysis was made with the Kaplan-Meier method and the Breslow test was used to test the statistical significance in the difference in survival between the two groups. The preoperatively patient characteristics were compared using a $\chi^2$ test.

Using the Cox proportional hazards regression method, the contribution of multiple covariates to survival was analysed and presented as HR with 95% CI and probability p. Besides an UPT, patient age and confirmed primary tumour type in biopsy were identified as relevant covariates for survival. Since we assumed that oncological advancement had improved patient survival in general in recent decades, decade of admission was included as a covariate in the model. Covariates were treated as categorical variables.

## Linkage

Each entry in Swespine is labelled with the PIN of the patient. The PIN was used to access the medical records including data on survival from the database at Uppsala University Hospital. Data from the medical records was manually entered into an anonymised database of the cases, stored in a secure location.

## RESULTS

Three hundred and ninety-three patients registered in Swespine met the inclusion criteria and were selected for

**Table 2** Distribution of primary tumours in the KPT and UPT groups

| Tumour type per group | KPT (as known before surgery) n | KPT (as known before surgery) % | UPT (after histopathology) n | UPT (after histopathology) % | P value |
|---|---|---|---|---|---|
| Prostate | 100 | 37 | 7 | 6 | <0.0001 |
| Blood | 34 | 13 | 43 | 35 | <0.0001 |
| Lung | 28 | 10 | 15 | 12 | 0.56 |
| Renal | 26 | 10 | 1 | 1 | 0.001 |
| Breast | 19 | 7 | 9 | 7 | 0.90 |
| Gastrointestinal | 16 | 6 | 10 | 8 | 0.40 |
| Thyroid | 4 | 2 | 1 | 1 | 0.59 |
| Others | 44 | 16 | 36 | 30 | 0.03 |
| Total | 271 | | 122 | | |

KPT, known primary tumour; UPT, unknown primary tumour.

**Table 3** Univariate and multivariate Cox regression analysis of covariates impacting survival after surgery

| Covariate | Categories | N deaths/ subjects | N person- months | Univariate | | Multivariate | |
|---|---|---|---|---|---|---|---|
| | | | | HR (95% CI) | P value | HR (95% CI) | P value |
| Primary tumour | | | | | | | |
| | Known | 257/271 | 5276 | 1.00 (Ref) | | 1.00 (Ref) | |
| | Unknown | 97/122 | 4784 | 0.58 (0.46 to 0.73) | <0.001 | 0.69 (0.53 to 0.90) | 0.005 |
| Age | | | | | | | |
| | <50 years | 15/22 | 1440 | 1.00 (Ref) | 0.24 | 1.00 (Ref) | <0.001 |
| | 50–64 years | 104/127 | 4132 | 1.97 (1.14 to 3.41) | 0.15 | 2.82 (1.62 to 4.90) | <0.001 |
| | 65–80 years | 192/201 | 4016 | 1.93 (1.13 to 3.30) | 0.16 | 3.28 (1.90 to 5.66) | <0.001 |
| | >80 years | 43/43 | 473 | 2.55 (1.40 to 4.64) | 0.002 | 5.55 (2.99 to 10.30) | <0.001 |
| Primary tumour in biopsy | | | | | | | |
| | Prostate | 106/107 | 1791 | 1.00 (Ref) | | 1.00 (Ref) | |
| | Blood | 49/77 | 449 | 0.30 (0.21 to 0.42) | <0.001 | 0.42 (0.28 to 0.62) | <0.001 |
| | Lung | 43/43 | 504 | 1.64 (1.15 to 2.35) | 0.007 | 2.53 (1.72 to 3.71) | <0.001 |
| | Renal | 25/27 | 453 | 0.95 (0.61 to 1.46) | 0.80 | 1.02 (0.65 to 1.59) | 0.94 |
| | Breast | 28/28 | 806 | 0.75 (0.49 to 1.14) | 0.18 | 0.96 (0.62 to 1.47) | 0.84 |
| | Gastrointestinal | 22/26 | 522 | 0.86 (0.54 to 1.36) | 0.52 | 1.50 (0.91 to 2.48) | 0.11 |
| | Thyroid | 5/5 | 204 | 0.59 (0.24 to 1.45) | 0.30 | 0.70 (0.28 to 1.72) | 0.44 |
| | Others | 76/80 | 1663 | 1.17 (0.87 to 1.57) | 0.30 | 1.61 (1.18 to 2.21) | 0.03 |
| Decade of surgery | | | | | | | |
| | 2000s | 185/196 | 4643 | 1.00 (Ref) | | 1.00 (Ref) | |
| | 2010s | 169/197 | 5417 | 0.94 (0.76 to 1.16) | 0.54 | 0.89 (0.71 to 1.11) | 0.30 |

further analysis. Two hundred and seventy were men, 123 were women and the mean age at the time of surgery was 67.3 years (range 23–91 years). Two hundred and seventy-one of the patients had a KPT verified in the medical records while 122 had an UPT at the time of surgery. There was a statistically significant lower percentage of men in the UPT group but no difference regarding age between the two groups. The level of missing data on smoking status was too high for any conclusion to be drawn (table 1).

**Survival**

The median estimated survival time after surgery for patients in the KPT group was 7.4 months (95% CI 6.0 to 8.7) and mean estimated survival was 21.6 months (95% CI 17.2 to 26.0). For patients in the UPT group, the median estimated survival was 15.6 months (95% CI 7.5 to 23.7) and mean estimated survival was 48.1 months (95% CI 37.3 to 59.0). The estimated 5-year survival was 9% in the KPT group and 28% in the UPT group.

**Table 4** Indications for surgery, as reported to Swespine. In the groups marked with (-) there were too few observations to calculate a p value

| Indication for surgery | KPT group n | KPT group % | UPT group n | UPT group % | P value |
|---|---|---|---|---|---|
| Neurological deficit (A) | 161 | 59 | 71 | 58 | 0.82 |
| Pain (B) | 42 | 16 | 12 | 10 | 0.13 |
| Progressive deformity (C) | 2 | 1 | 3 | 3 | 0.16 |
| A+B | 27 | 10 | 17 | 14 | 0.25 |
| A+C | 0 | 0 | 2 | 2 | (-) |
| B+C | 2 | 1 | 1 | 1 | 0.93 |
| A+B+C | 6 | 2 | 0 | 0 | (-) |
| Missing data | 31 | 11 | 16 | 13 | 0.64 |
| Total | 271 | | 122 | | |

KPT, known primary tumour; UPT, unknown primary tumour.

**Table 5** Types of surgery, as reported to Swespine

| Type of surgery | KPT group n | KPT group % | UPT group n | UPT group % | Missing data n | P value |
|---|---|---|---|---|---|---|
| Posterior decompression | 219 | 81 | 97 | 80 | 6 | 0.76 |
| Posterior implant | 213 | 79 | 94 | 77 | 8 | 0.73 |
| Anterior decompression | 35 | 13 | 19 | 16 | 30 | 0.49 |
| Anterior implant | 42 | 15 | 22 | 18 | 17 | 0.53 |

KPT, known primary tumour; UPT, unknown primary tumour.

Estimated 10-year survival was 5% in the KPT group and 18% in the UPT group. After comparing the two groups with a Breslow test (generalised Wilcoxon), the p value was 0.001 and the difference in survival was considered as statistically significant. As of 25 January 2021, 39 of the patients (10%) in the cohort were alive (figure 2).

### Tumour types

The most common primary tumour in the KPT group was prostate cancer, followed by lung cancer and blood, which in most cases means myeloma/plasmacytoma or lymphoma. In the UPT group, blood was the most common specified origin followed by lung cancer. The group 'others' is a heterogeneous group where still unidentified tumours, even after the postoperative histopathology, account for the majority. Cases with only non-specific pathology report results such as 'squamous cells' are included here. The difference in distribution of primary tumours between the KPT and UPT groups reached statistical significance (p<0.05) in four categories (table 2).

Tumours originating from the blood had the lowest HR of the investigated tumour types, while lung tumours had the highest HR (table 3).

### Indications for surgery

Neurological deficit was the most common indication for surgery while posterior decompression with posterior implants (posterior stabilisation) was the most common surgical procedure in both groups. The distributions of indications for surgery and types of surgery were not statistically significant between the two groups (tables 4 and 5).

### DISCUSSION

This is, to the best of our knowledge, one of the largest single-centre cohorts with surgically treated patients in recent time. The single-centre cohort study design facilitates access to medical records and reliable survival data, thanks to the linkage to the Swedish Population Register.

The distribution of primary tumours in this study is different from other studies. The prospective AO Spine study by Fehlings *et al* published in 2016 excluded patients with known haematological malignancies, but had three other tumour groups (lung, kidney, breast) ahead of prostate, which is the most common KPT in our Swedish material.[2] One possible explanation might be geographical, as the study by Wang *et al* performed in neighbouring Denmark has prostate cancer as the most common KPT followed by breast and kidney.[14]

There are several possible explanations to the longer survival in the UPT group compared with the KPT group. First, the distribution of malignancies is different. In the UPT group, there is a higher rate of myeloma and lymphoma, cancer types known to respond very well to radiation. If the primary tumour had been known at the time of surgery, acute radiation rather than decompressive surgery would probably have been an option for some of those patients. The lack of availability of emergency radiation treatment during weekends at referring hospitals is another possible reason for the high amount of surgically treated cases in the group, underlining the need for access to acute radiation, preferably within 24 hours.[16]

Another explanation for the longer survival in the UPT group is that the tumour is treatment naïve and the field of oncological treatment is still open. Furthermore, the state of health in patients with spinal metastasis as IMM does not per definition cause an earlier urgent investigation of a possible disseminated cancer, which suggests a good general health status.

One possible weakness of this study is the lack of a control group of non-surgically treated patients, especially since there is a possible selection bias among patients deemed fit to undergo spine surgery. Another potential field of improvement is reliable reporting of other outcomes, such as complications and the impact of health-related quality of life after surgery.

Nevertheless, the findings of our study suggest that patients with spinal metastatic disease as IMM should not be withheld from surgical treatment based on the fact that the primary tumour is unknown. The decision on whether to perform extensive spine surgery or not should be based on other factors, underlining the need for reliable tools for decision-making. We suggest that further research in the field of surgical treatment of spinal metastatic should focus on other outcomes than survival, such as quality of life and complications after treatment.

### Generalisability

The results of this study do not apply to all patients with spinal metastatic disease, due to the heterogeneity between different types of cancer.

**Contributors** CC designed the study, drafted the manuscript and made the statistical analyses. CO was involved in the study design and reviewed the manuscript. YR was involved in the study design, wrote the ethical board application and reviewed the manuscript.

**Funding** The authors have not declared a specific grant for this research from any funding agency in the public, commercial or not-for-profit sectors.

**Competing interests** CO reports personal fees from Johnson & Johnson (paid speaker on course) outside the submitted work.

**Patient and public involvement** Patients and/or the public were not involved in the design, or conduct, or reporting, or dissemination plans of this research.

**Patient consent for publication** Not required.

**Ethics approval** This study was approved by the Regional Ethical Review Board in Uppsala (reference 2012/133).

**Provenance and peer review** Not commissioned; externally peer reviewed.

**Data availability statement** Data may be obtained from a third party and are not publicly available. Source data provided by Swespine and from medical records stored by Region Uppsala. Swespine is available to researchers subject to application (http://www.swespine.se).

**ORCID iDs**
Christian Carrwik http://orcid.org/0000-0002-3092-8139
Yohan Robinson http://orcid.org/0000-0002-2724-6372

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
