## [Reviewer comments · BMJ Open]

ARTICLE DETAILS

TITLE (PROVISIONAL)	Does knowledge of the primary tumour affect survival after surgery for spinal metastatic disease? A retrospective longitudinal cohort study
AUTHORS	Carrwik, Christian; Olerud, Claes; Robinson, Yohan

VERSION 1 – REVIEW

REVIEWER	Reiner, Anne Memorial Sloan Kettering Cancer Center
REVIEW RETURNED	16-Apr-2021

GENERAL COMMENTS	The authors add to prior conflicting literature of unknown/known primary cancer site and association with overall survival following surgery for spinal mets. The retrospective study included all 18+ participants undergoing spine mets surgery from 2006-2016 in Swespine. The authors report that participants with unknown primary tumor had longer OS compared with their counterparts with known primary tumor. The biggest weakness is the lack of information on participants with spine mets who do not undergo surgery during the same time period. The authors clearly note this and the bias that patients selected to undergo surgery will most likely have better performance status. Thus, these results are not akin to a randomized control trial. However, the conclusion is fair that unknown primary tumor should not be the only reason to preclude spine mets. Some comments: 1. What do the two numbers mean in each cell of the known smoker row of Table 1?2. Why not combine Tables 2 and 3 in the same vein as Table 4 so that the reader can compare eventual primary tumor types by known/unknown primary tumor status at time of surgery?3. The authors note that the 'others' category in Tables 2 and 3 is a heterogeneous group where unidentified tumors account for the majority. This seems to be in direct opposition to the fact that participants in Table 2 are with known primary tumor.4. Can the authors include p-values for Tables 2/3 (combined) and Tables 4 and 5/5. Because there is a highly statistically significant association between eventual primary tumor status and UPT/KPT, can the authors perform multivariable Cox regression to examine this? The outcome would be time to death (or censor for those still alive) and
---

	variables in the model would include UPT/KPT, sex, and eventual primary tumor status. Perhaps primary tumor status is correlated with baseline performance status? A model with UPT/KPT and sex would also be of interest, with OS as outcome. 6. The total and missing data proportion is incorrect for UPT in Table 4.
--	---

REVIEWER	Agel, Julie Harborview Medical Center, Orthopaedics
REVIEW RETURNED	16-Apr-2021

GENERAL COMMENTS	This is a nice straightforward project. I have few specific comments. Please provide a little information as to what the time line was for patients. How many had five years vs 10 years. Line 21 on page 13 requires an additional word. Your flow chart boxes should be based on enrollment numbers not steps - Somewhere please give the absolute numbers of patients who expired. (and if all deaths were attributable to tumors)
--

VERSION 1 – AUTHOR RESPONSE

1. What do the two numbers mean in each cell of the known smoker row of Table 1?

This is now clarified in Table 1.

2. Why not combine Tables 2 and 3 in the same vein as Table 4 so that the reader can compare eventual primary tumor types by known/unknown primary tumor status at time of surgery?

Tables 2 and 3 are now combined.

3. The authors note that the 'others' category in Tables 2 and 3 is a heterogeneous group where unidentified tumors account for the majority. This seems to be in direct opposition to the fact that participants in Table 2 are with known primary tumor.

The term “unidentified” has now been explained in more detail – it is not the same as unknown primary tumour.

4. Can the authors include p-values for Tables 2/3 (combined) and Tables 4 and 5/

P-values are now included.

5. Because there is a highly statistically significant association between eventual primary tumor status and UPT/KPT, can the authors perform multivariable Cox regression to examine this? The outcome would be time to death (or censor for those still alive) and variables in the model would include UPT/KPT, sex, and eventual primary tumor status. Perhaps primary tumor status is correlated with baseline performance status? A model with UPT/KPT and sex would also be of interest, with OS as outcome.

A cox regression analysis with primary tumours and other factors is now included in the paper. However, we do not see a model using sex as a variable as useful, because of the sex-specific distribution of the large groups of prostate and breast cancer.

6. The total and missing data proportion is incorrect for UPT in Table 4.

This is now corrected.

Reviewer: 2

Ms. Julie Agel, Harborview Medical Center

Comments to the Author:

This is a nice straightforward project. I have few specific comments. Please provide a little information as to what the time line was for patients. How many had five years vs 10 years.

This is now included in the "Survival" part of the results.

Line 21 on page 13 requires an additional word.

Corrected.

Your flow chart boxes should be based on enrollment numbers not steps.

The flow chart is now updated.

Somewhere please give the absolute numbers of patients who expired. (and if all deaths were attributable to tumors)

The absolute number of deceased patients is now included in the "Survival" part of the results. Unfortunately, this study had no access to cause of death data.

VERSION 2 – REVIEW

REVIEWER	Reiner, Anne Memorial Sloan Kettering Cancer Center
REVIEW RETURNED	28-Jun-2021

GENERAL COMMENTS	Thank you for the responses. Two remaining comments: 1) I still do not understand the statement on page 11: "The group "others" is a heterogenous group where still unidentified tumours, even after histopathology, account for the majority." Do the authors mean the "others" group consists of mostly unidentified tumors in the UPT group? Is the "others" group for the KPT group all identified? 2. The authors have added the following statement on page in response to another reviewer: "Regarding true survival data, 58 of
---

	the patients (15%) survived at least five years after surgery and ten years after surgery 15 patients (4%) were still alive. As of January 25th 2021, 39 of the patients (10%) in the cohort were alive". It would be more appropriate to provide Kaplan-Meier based 5- and 10-year survival estimates with corresponding 95% confidence intervals.
--	--

VERSION 2 – AUTHOR RESPONSE

1) I still do not understand the statement on page 11: "The group "others" is a heterogenous group where still unidentified tumours, even after histopathology, account for the majority." Do the authors mean the "others" group consists of mostly unidentified tumors in the UPT group? Is the "others" group for the KPT group all identified?

We certainly understand the need for clarification here. In the UPT group, the group "others" consists of tumour types with low frequencies in the study group. However, in 24 of the cases in the UPT group, no definitive result was given even after histopathology. "Unknown origin" and non-specific results such as "squamous epithelial cells" are included here. This has now been updated in the main text.

Regarding the "others" group in the KPT group, these are registered in Swespine as having a known primary tumour not fitting in any of the seven specified categories and thus registered in the eighth category as "others". We have not dug deeper on how the diagnosis was established in the first place, but from a clinical point of view the surgeon treated a patient with a KPT rather than an UPT.

2. The authors have added the following statement on page in response to another reviewer: "Regarding true survival data, 58 of the patients (15%) survived at least five years after surgery and ten years after surgery 15 patients (4%) were still alive. As of January 25th 2021, 39 of the patients (10%) in the cohort were alive". It would be more appropriate to provide Kaplan-Meier based 5- and 10-year survival estimates with corresponding 95% confidence intervals.

The estimated 5- and 10-year survival estimates for the KPT and UPT groups are now included in the manuscript, as suggested. The Kaplan-Meier procedure in SPSS is unable to produce confidence intervals for each time point. The 95% CI can be calculated manually but the method is not recommended in small samples such as ours, according to the SPSS manual and other statistical sources as it can produce confidence intervals outside the 0-1 interval. We seriously considered including 95% CI, but due to sample size and the above mentioned obvious uncertainties with the calculation of 95% CI in our database we refrained from doing so.